# Emergent Visual Representations through Unsupervised Spiking Networks with Synaptic Pruning

Di Hong [1]   Dazhong Rong [1 2]   Yueming Wang [1 2]

## Abstract

Recent work has shown that brain-aligned visual representations can emerge even in randomly initialized, high-dimensional neural networks, suggesting that cortical representations may be discovered rather than fully learned through task optimization. However, how such latent brain-relevant representations are stabilized and refined during development remains unclear. Motivated by this perspective and by neuroscientific evidence of activity-dependent synaptic pruning, we study how brain-aligned representations can emerge and be refined from high-dimensional unsupervised spiking systems. We propose a biologically grounded deep SNN that integrates unsupervised learning with developmental pruning dynamics. Starting from an overcomplete spiking architecture, the model self-organizes through sensory-driven activity while selectively eliminating weak or redundant synapses, progressively yielding compact and informative representations. Without using labels, the resulting network forms hierarchical visual representations that strongly align with neural responses across multiple areas of the mouse and macaque visual cortex, outperforming supervised and unsupervised ANN and SNN baselines. Synaptic pruning further improves alignment and robustness under noisy and few-shot recognition settings. By unifying high-dimensional unsupervised spiking representations with activity-dependent synaptic pruning, this work provides a computational account of developmental refinement in visual cortex and bridges recent findings on emergent brain alignment in random networks with biologically grounded models of representation learning.

[1]College of Computer Science and Technology, Zhejiang University, Hangzhou, China [2]NANHU Brain-Computer Interface Institute, Hangzhou, China. Correspondence to: Yueming Wang <ymingwang@zju.edu.cn>.

*Proceedings of the $43^{rd}$ International Conference on Machine Learning*, Seoul, South Korea. PMLR 306, 2026. Copyright 2026 by the author(s).

## 1. Introduction

The primate visual system transforms raw sensory input into meaningful object representations through a hierarchically organized cortical pathway known as the *ventral visual stream* (Movshon et al., 1978; Malach et al., 1995). Early visual areas (e.g., V1) encode low-level features such as edges and local contrast (Carandini et al., 2005), while intermediate regions (V2, V3, V4) progressively integrate these features into more complex structures (Gallant et al., 1996; Schmolesky et al., 1998; Freeman & Simoncelli, 2011; Lennie & Movshon, 2005; Brincat & Connor, 2004). At the apex of this hierarchy, inferotemporal (IT) cortex represents object identity in a linearly decodable and transformation-tolerant manner (Yamane et al., 2008). This organization has long inspired artificial neural networks (ANNs), whose layered architectures mirror key computational motifs of biological vision (LeCun et al., 1995; Cadena et al., 2019).

Over the past decade, *task-optimized ANNs*, particularly deep convolutional neural networks (CNNs), have become quantitatively accurate models of the ventral visual stream. Although not fitted directly to neural recordings, networks optimized for ecologically relevant visual tasks exhibit internal representations that align remarkably well with neural activity across multiple cortical areas (Yamins et al., 2014; Yamins & DiCarlo, 2016). This success established task-driven optimization as a dominant paradigm in computational neuroscience, suggesting that functional objectives can implicitly shape brain-like representations.

Recent evidence, however, challenges the view that extensive task optimization is strictly necessary for brain alignment. Notably, high-dimensional convolutional architectures with minimal or no task training can already exhibit substantial correspondence with primate visual cortex when appropriate architectural inductive biases are present (Kazemian et al., 2025). These findings suggest that brain-relevant representations may, to a significant extent, be *discovered* within overcomplete feature spaces rather than fully *learned* through supervision. This perspective raises a fundamental question: *how are such latent brain-aligned representations stabilized, refined, and retained during biological development?*

Neuroscience points to a compelling mechanism. During development, cortical circuits initially exhibit exuberant synaptic connectivity, followed by prolonged periods of *activity-dependent synaptic pruning* that selectively eliminate weak or redundant connections (Paolicelli et al., 2011; Sakai, 2020). Rather than merely compressing networks, pruning is thought to refine representational structure, improve efficiency, and stabilize functionally relevant subcircuits. From this perspective, synaptic pruning can be viewed as a biological process that distills informative representations from an initially high-dimensional substrate, closely paralleling recent computational observations of representational distillation in random networks.

From a learning standpoint, unsupervised objectives are strongly motivated by evidence that contrastive learning can yield neural predictivity comparable to or exceeding that of supervised models, even under realistic, label-sparse sensory experience (Zhuang et al., 2021). Complementarily, large-scale neural similarity analyses show that deep spiking neural networks (SNNs) achieve systematically stronger alignment with mouse and macaque visual cortex than architecture-matched CNNs, highlighting the importance of spiking dynamics for modeling cortical computation (Huang et al., 2023). Together, these findings motivate unsupervised SNNs as a biologically grounded framework for studying how high-dimensional visual representations emerge and are refined over development.

In this work, we propose a deep *unsupervised spiking neural network* equipped with *activity-dependent synaptic pruning* as a computational model of visual cortical development. Starting from an overcomplete spiking architecture, the network self-organizes through sensory-driven activity and progressively refines its connectivity via pruning. We show that this process yields compact, hierarchical representations that align closely with neural responses across mouse and macaque visual cortex and support robust object recognition under noise and limited supervision.

This work makes the following contributions:

- We propose a **deep unsupervised spiking neural network with activity-dependent synaptic pruning** as a biologically grounded model of visual cortical refinement, unifying unsupervised representation learning with developmental structural dynamics.

- We demonstrate that, starting from an **overcomplete, high-dimensional spiking architecture**, unsupervised learning combined with pruning yields **compact and hierarchical representations** that closely align with neural responses across multiple areas of the mouse and macaque visual cortex.

- Through extensive model-to-brain evaluations, we show that **unsupervised spiking representations consistently outperform supervised and non-spiking baselines** in neural predictivity, and that synaptic pruning further and systematically enhances this alignment.

- We show that the same representations support **robust downstream behavior**, achieving strong performance under noisy and few-shot recognition settings, linking neural alignment with functional generalization.

- Conceptually, our results provide a **computational bridge between emergent brain alignment in random high-dimensional networks and developmental refinement mechanisms**, suggesting synaptic pruning as a principled process for stabilizing and distilling brain-aligned representations.

## 2. Related Work

Understanding how artificial and biological neural systems converge in their visual representations is a central problem in computational neuroscience. Prior work can be broadly grouped into task-driven ANN models, architecture-driven accounts emphasizing inductive biases, and biologically grounded spiking models that explicitly capture neural dynamics and development.

### 2.1. Task-Driven and Unsupervised ANNs as Models of the Visual Pathway

Task-optimized ANNs, particularly deep convolutional networks trained for object recognition, have emerged as powerful models of the primate ventral visual stream. Such networks reproduce key aspects of cortical hierarchy, from early visual areas to inferotemporal cortex, and their internal representations align with neural responses despite not being fitted directly to neural data (Yamins et al., 2014; Yamins & DiCarlo, 2016; Cadena et al., 2019). Extensions incorporating recurrence and feedback further improve temporal and behavioral correspondence (Kubilius et al., 2019; Kar et al., 2019). More recently, unsupervised and contrastive learning approaches have demonstrated that high neural predictivity can be achieved without category supervision, providing a more developmentally plausible account of ventral stream learning (Zhuang et al., 2021; Rong et al., 2025). Nevertheless, conventional ANNs rely on non-spiking units and static connectivity, limiting their ability to model temporal coding and developmental circuit refinement.

### 2.2. Architecture-Driven Emergence of Brain-Aligned Representations

Recent work has challenged the necessity of extensive task optimization by showing that high-dimensional convolutional architectures with minimal or no training can already exhibit substantial alignment with primate visual cor-

tex, provided that appropriate architectural inductive biases are present (Kazemian et al., 2025). These findings suggest that brain-relevant representations may be *discovered* within overcomplete feature spaces rather than fully *learned* through supervision. However, such models typically rely on linear readouts from random networks and do not address how initially overcomplete representations are refined, stabilized, or selectively retained during biological development.

### 2.3. Spiking Neural Networks and Biologically Grounded Visual Models

Spiking neural networks provide a biologically grounded alternative by modeling neurons as event-driven units that communicate via discrete spikes, capturing key aspects of cortical temporal dynamics (Neftci et al., 2019). Recent studies show that deep SNNs can match or exceed the neural representational similarity of architecture-matched CNNs when evaluated against mouse and macaque visual cortex data (Huang et al., 2023). However, most existing SNN models rely on supervised training with large labeled datasets and treat structural plasticity, when present, as an optimization heuristic rather than a principled developmental mechanism. In contrast, our work integrates unsupervised spiking computation with activity-dependent synaptic pruning, directly addressing how brain-aligned representations can emerge and be refined within a developmentally plausible computational framework.

More broadly, recent theoretical and empirical work has emphasized that high-dimensional neural representations can contain latent structure aligned with biological data even prior to learning, raising questions about how such structure is selectively stabilized and functionally organized. While architectural inductive biases and random feature models demonstrate the existence of brain-aligned subspaces, they do not explain how biological systems transform these overcomplete representations into efficient and robust circuits. Developmental mechanisms such as synaptic pruning, which are ubiquitous in cortex but largely absent from current modeling approaches, offer a potential bridge between representational discovery and functional specialization. Our work builds on these insights by explicitly modeling how unsupervised spiking dynamics and activity-dependent pruning jointly refine high-dimensional representations into stable, brain-aligned visual hierarchies.

## 3. Methods

### 3.1. Spiking Neuron Model

We employ Leaky Integrate-and-Fire (LIF) neurons to model the spatio-temporal dynamics of biological neurons in the ventral visual stream. The LIF neuron integrates input spike information over time, accumulating it in its membrane po-

tential. This potential gradually decays, allowing controlled forgetting of less relevant information. The membrane potential $u(t)$ of a LIF neuron at timestep $t$ evolves as:

$$u_i(t) = \lambda u_i(t-1) + \sum_j \omega_{ij} o_j(t) - v_{th} o_i(t-1) \quad (1)$$

where $\lambda \in (0,1)$ is the leaking factor, controlling the decay rate of the membrane potential (biological analogy: ion channel leakage), $\omega$ denotes the synaptic weight of the presynaptic neuron, and $v_{th}$ represents the firing threshold. When the membrane potential $u(t)$ surpass the voltage threshold $v_{th}$, the neuron omits a spike. Then, the membrane potential of the omitted neuron reset to zero (hard reset), the mechanism can be described as

$$s(t) = \frac{u(t)}{v_{th}} - 1, \quad o(t) = \begin{cases} 1, & if \ s(t) > 0, \\ 0, & otherwise \end{cases} \quad (2)$$

Although temporal simulation in LIF-based SNNs introduces additional computational cost compared to standard ANN inference, the overhead is mitigated in our framework by employing a small number of timesteps ($T = 4$). Furthermore, the proposed synaptic pruning mechanism removes redundant connections and partially offsets the computational complexity introduced by temporal dynamics, improving overall efficiency while preserving biologically aligned representations.

### 3.2. Unsupervised Contrastive Learning Framework

Biological vision systems learn robust representations through temporally structured sensory experience rather than labeled data. To emulate the developmental learning process observed in biological vision systems, we designed a spike-based contrastive learning framework that integrates temporal dynamics with self-supervised objectives. As shown in Figure 1, this framework begins by simulating natural variations in visual input through stochastic transformations $\tau(\cdot; \rho) : \mathbb{R}^d \to \mathbb{R}^d$, including random crops parameterized by spatial coordinates $\rho = [x, y, scale]$, color jitter (adjusting brightness and contrast), and horizontal flipping. These augmentations mimic the diverse sensory experiences encountered during biological development, enabling the network to learn invariant representations without labeled data.

The core of the framework is a temporal contrastive loss that operates on spike trains generated over $t$ timesteps. At each timestep, spike trains are converted into continuous embeddings via temporal averaging of membrane potentials (or spike counts), yielding $Z_{i,t}$. For a batch of $N$ images, augmented pairs $(X^A, X^B)$ are encoded in terms of spikes and fed forward into an encoder network $f(\cdot)$ to obtain the representation $h$, which is then passed through a projection head $g(\cdot)$ to obtain $Z_{i,t}^{A/B} \in \mathbb{R}^d$. The projection head $g(\cdot)$

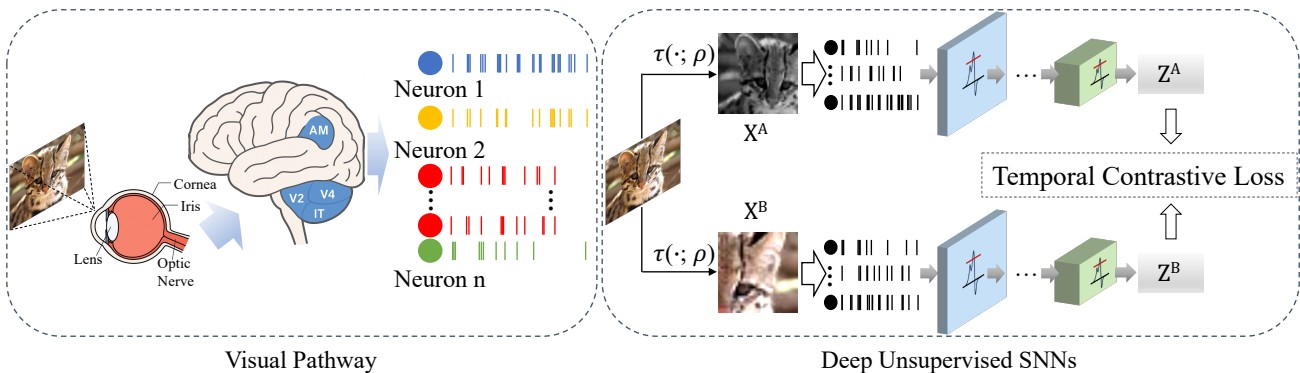

*Figure 1.* The comparative framework of the biological visual pathway and deep unsupervised spiking neural networks.

is implemented as a lightweight multilayer perceptron and is discarded after training. The loss function maximizes similarity between positive pairs (augmented views of the same image) while minimizing similarity to negative pairs (different images), formalized as:

$$\mathcal{L} = -\frac{1}{T} \sum_{i \in 1}^{N} \sum_{t=1}^{T} log \frac{exp(sim(Z_{i,t}^A, Z_{i,t}^B)/\tau)}{\sum_{j \neq i} exp(sim(Z_{i,t}^A, Z_{j,t}^B)/\tau)} \quad (3)$$

where $sim(a,b) = a^\top b/||a||||b||$ computes cosine similarity, and $\tau$ controls the sharpness of the similarity distribution. Following standard practice, we remove the projection head $g(\cdot)$ after training and retain only the encoder $f(\cdot)$ and representations $h$ for downstream tasks.

Biologically, this framework aligns with two key mechanisms of developmental plasticity, including temporal dynamics and self-supervised learning. Temporal dynamics ensures that spike timing encodes stimulus variations, analogous to latency coding in retinal ganglion cells, where earlier spikes convey salient features. Self-supervised learning eliminates dependency on explicit labels, mirroring experience-driven refinement in juvenile primates, where visual circuits mature through exposure to natural scene statistics rather than supervised feedback.

### 3.3. Synaptic Pruning

To bridge artificial models with biological developmental mechanisms, we introduce a synaptic pruning framework that removes redundant connections while preserving functionally essential pathways. By applying static pruning at varying ratios, we emulate the developmental refinement observed in the biological visual pathway. This approach extends traditional ANN pruning methods by incorporating temporal cosine similarity—a metric that quantifies directional alignment between original and pruned synaptic weight vectors $\omega_t$ and $\omega_t'$ at timestep $t$, defined as:

$$S_C^t(\omega_t, \omega_t') := cos(\phi_{\omega_t, \omega_t'}) = \frac{\langle \omega_t, \omega_t' \rangle}{||\omega_t||_2 \, ||\omega_t'||_2}, \quad (4)$$

synapses with consistently low temporal cosine similarity across timesteps are removed to achieve a target pruning ratio. Although pruning is applied statically after training, the pruning criterion is derived from activity-dependent temporal statistics, thereby approximating developmental synaptic refinement.

## 4. Experiments

**Implementation Details.** Our proposed methods consist of two variants built on SNN-based ResNet backbones. **UL SNN** is trained using an unsupervised spike-based contrastive objective, enabling biologically aligned representation learning without category labels. Building on this model, **Pr. UL SNN** further incorporates activity-dependent synaptic pruning, introducing a biologically inspired structural refinement mechanism that progressively shapes compact and informative representations.

All spiking neural network models are simulated for a fixed number of timesteps ($T = 4$). For unsupervised training (UL ANN and UL SNN), we apply a consistent set of data augmentations, including random cropping, horizontal flipping, color jitter, and grayscale conversion, following standard practice in contrastive learning.

**Baseline Methods.** Throughout the experiments, we use the following shorthand: **SL ANN** denotes conventional ResNet models trained with supervised image classification; **UL ANN** refers to the same architectures pretrained using SimCLR (Chen et al., 2020) (Zhuang et al., 2021); **SL SNN** denotes SNN-based ResNet models trained with supervised learning; and **SEW SNN** refers to SEW-ResNet (Fang et al., 2021), a supervised spiking variant with strong neural alignment (Huang et al., 2023).

**Evaluation Metrics.** We adopt a BrainScore-style evaluation pipeline (Schrimpf et al., 2018) to quantify *neural predictivity*, measuring alignment between model representations and neural responses along the ventral visual

*Table 1.* Brain scores on the Allen Brain Mouse dataset across different methods (CIFAR-10 pretraining).

| Metrics | Methods | Allen Brain Mouse (CIFAR-10) | |
| --- | --- | --- | --- |
| | | ResNet18 | ResNet34 |
| SVCCA | SL ANN | 0.6075 | 0.5914 |
| | UL ANN | 0.6046 | 0.5955 |
| | SL SNN | 0.5680 | 0.5680 |
| | SEW | 0.6065 | 0.5821 |
| | **UL SNN** | 0.6294 | 0.6104 |
| | **Pr. UL SNN**$^*$ | **0.6308** | **0.6141** |
| RSA | SL ANN | 0.1852 | 0.1808 |
| | UL ANN | 0.1849 | 0.1868 |
| | SL SNN | 0.1601 | 0.2259 |
| | SEW | 0.2603 | 0.2763 |
| | **UL SNN** | 0.4279 | 0.4132 |
| | **Pr. UL SNN**$^*$ | **0.4608** | **0.4367** |

*Table 2.* Brain scores on the Allen Brain Mouse dataset across different methods (Tiny-ImageNet pretraining).

| Metrics | Methods | Allen Brain Mouse (Tiny-ImageNet) | |
| --- | --- | --- | --- |
| | | ResNet18 | ResNet34 |
| SVCCA | SL ANN | 0.5706 | 0.5987 |
| | UL ANN | 0.5615 | 0.6012 |
| | SL SNN | 0.5788 | 0.5721 |
| | SEW | 0.5738 | 0.6100 |
| | **UL SNN** | 0.5961 | 0.6135 |
| | **Pr. UL SNN**$^*$ | **0.5990** | **0.6179** |
| RSA | SL ANN | 0.1703 | 0.1851 |
| | UL ANN | 0.2228 | 0.2450 |
| | SL SNN | 0.1016 | 0.2298 |
| | SEW | 0.2330 | 0.1991 |
| | **UL SNN** | 0.3786 | 0.3881 |
| | **Pr. UL SNN**$^*$ | **0.4156** | **0.3995** |

*Table 3.* Brain scores on the Macaque Face dataset (CIFAR-10 pretraining).

| Metrics | Methods | Macaque Face (CIFAR-10) | |
| --- | --- | --- | --- |
| | | ResNet18 | ResNet34 |
| SVCCA | SL ANN | 0.4393 | 0.4383 |
| | UL ANN | 0.4385 | 0.4371 |
| | SL SNN | 0.3949 | 0.3887 |
| | SEW | 0.4578 | **0.4576** |
| | **UL SNN** | 0.4576 | 0.4464 |
| | **Pr. UL SNN**$^*$ | **0.4687** | 0.4533 |
| TSVDR | SL ANN | 0.4457 | 0.4449 |
| | UL ANN | 0.4455 | 0.4449 |
| | SL SNN | 0.3908 | 0.3830 |
| | SEW | 0.4439 | 0.4527 |
| | **UL SNN** | 0.4533 | 0.4536 |
| | **Pr. UL SNN**$^*$ | **0.4636** | **0.4615** |

stream. Following (Huang et al., 2023), we use three complementary metrics—singular vector canonical correlation analysis (SVCCA), truncated singular value decomposition regression (TSVDR), and representational similarity analysis (RSA)—to capture both linear and representational-geometry correspondence.

Models are evaluated on three benchmark neural datasets (Allen Brain Mouse, Macaque Face, and Macaque Synthetic), spanning early to high-level visual areas across species. Unless otherwise noted, all analyses are conducted on fully specified pretrained models with fixed architectures, and results for both UL SNN and its pruned variant are reported; additional implementation details and ablation studies are provided in the Appendix.

### 4.1. Capturing Brain-like Representations with Unsupervised SNNs and Pruned Variants

We first compare supervised and unsupervised ANN/SNN models, together with our UL SNN and Pr. UL SNN, on neural predictivity across three datasets: Allen Brain Mouse, Macaque Face, and Macaque Synthetic. Unless otherwise noted, Pr. UL SNN$^*$ denotes the pruned UL SNN at the pruning ratio selected based on validation performance.

**Allen Brain Mouse dataset.** On the mouse visual cortex data, both UL SNN and Pr. UL SNN exhibit stronger representational alignment than supervised baselines. For example, with CIFAR-10 pretraining and a ResNet18 backbone (Table 1), UL SNN improves SVCCA from 0.6075 (SL ANN) and 0.6065 (SEW SNN) to 0.6294, while pruning further raises it to 0.6308. The gains are even more pronounced for RSA, where UL SNN reaches 0.4279 and Pr. UL SNN reaches 0.4608, substantially exceeding both ANN and supervised SNN baselines. Similar trends hold under Tiny-ImageNet pretraining (Table 2). These results suggest that unsupervised spiking computation, especially

when combined with synaptic pruning, more faithfully captures the stimulus-evoked response geometry of early and intermediate mouse visual areas.

**Macaque Face dataset.** On the macaque face-selective regions, Pr. UL SNN achieves the highest neural predictivity among all evaluated models across both CIFAR-10 and Tiny-ImageNet pretraining (Tables 3 and 4). UL SNN already matches or surpasses supervised SNNs (SL SNN, SEW SNN), and pruning provides consistent additional improvements in both SVCCA and TSVDR. Because this dataset targets high-level face patches analogous to IT cortex, these results indicate that unsupervised spiking learning, refined by activity-dependent pruning, better captures transformation-tolerant object encoding associated with higher-order visual processing.

**Macaque Synthetic dataset.** On the macaque synthetic stimuli, UL SNN and Pr. UL SNN maintain competitive or superior performance across backbones and metrics. With CIFAR-10 pretraining (Table 5), Pr. UL SNN achieves the

*Table 4.* Brain scores on the Macaque Face dataset (Tiny-ImageNet pretraining).

| Metrics | Methods | Macaque Face (Tiny-ImageNet) | |
|---|---|---|---|
| | | ResNet18 | ResNet34 |
| SVCCA | SL ANN | 0.4241 | 0.4390 |
| | UL ANN | 0.4093 | 0.4402 |
| | SL SNN | 0.4229 | 0.4093 |
| | SEW | 0.4366 | 0.4384 |
| | **UL SNN** | 0.4396 | 0.4409 |
| | **Pr. UL SNN*** | **0.4559** | **0.4549** |
| TSVDR | SL ANN | 0.4353 | 0.4455 |
| | UL ANN | 0.4046 | 0.4458 |
| | SL SNN | 0.4285 | 0.4125 |
| | SEW | 0.4295 | 0.4380 |
| | **UL SNN** | 0.4499 | 0.4539 |
| | **Pr. UL SNN*** | **0.4509** | **0.4591** |

*Table 5.* Brain scores on the Macaque Synthetic dataset (CIFAR-10 pretraining).

| Metrics | Methods | Macaque Synthetic (CIFAR-10) | |
|---|---|---|---|
| | | ResNet18 | ResNet34 |
| SVCCA | SL ANN | 0.3259 | 0.3098 |
| | UL ANN | 0.3270 | 0.3094 |
| | SL SNN | **0.3461** | 0.2871 |
| | **UL SNN** | 0.3194 | 0.3191 |
| | **Pr. UL SNN*** | 0.3226 | **0.3520** |
| TSVDR | SL ANN | 0.3125 | 0.3583 |
| | UL ANN | 0.3103 | 0.3470 |
| | SL SNN | 0.3499 | 0.2802 |
| | **UL SNN** | 0.3368 | 0.3284 |
| | **Pr. UL SNN*** | **0.3536** | **0.3619** |

highest TSVDR scores for both ResNet18 (0.3536) and ResNet34 (0.3619), and yields the strongest SVCCA for ResNet34 (0.3520). Under Tiny-ImageNet pretraining (Table 6), UL SNN and Pr. UL SNN achieve the best SVCCA values, while TSVDR remains close to the strongest ANN baseline. This robustness across diverse, shape-driven stimuli suggests that the proposed models generalize well to abstract feature spaces linked to mid- and high-level cortical computation.

Across species, architectures, and stimulus domains, UL SNN and its pruned variant consistently enhance neural predictivity relative to supervised ANNs and SNNs. Taken together, these results support three main observations: (1) unsupervised learning suffices to drive hierarchical representational development in a way that aligns with early-life visual experience; (2) spiking dynamics provide a more faithful substrate for brain-like temporal and population codes than standard rate-based ANNs; and (3) activity-dependent synaptic pruning yields further gains in biological alignment, consistent with the developmental refinement of cortical circuits. Combining unsupervised sensory learning with

*Table 6.* Brain scores on the Macaque Synthetic dataset (Tiny-ImageNet pretraining, ResNet18).

| Methods | Macaque Synthetic (Tiny-ImageNet) | |
|---|---|---|
| | SVCCA (ResNet18) | TSVDR (ResNet18) |
| SL ANN | 0.3101 | **0.4048** |
| UL ANN | 0.3271 | 0.3926 |
| SL SNN | 0.3021 | 0.3672 |
| **UL SNN** | 0.3445 | 0.3556 |
| **Pr. UL SNN*** | **0.3590** | 0.3668 |

structural refinement thus offers a biologically grounded route to emergent, cortex-like visual representations.

### 4.2. Layer–Area Correspondence in the AM Face Patch

To further examine whether unsupervised spiking learning gives rise to biologically meaningful hierarchical representations, we analyze the layer-wise correspondence between network activations and neural responses recorded from the macaque AM face patch. This cortical region lies at the apex of the ventral visual stream and is known to encode identity-invariant facial information despite changes in viewpoint, illumination, and expression.

Across all three representational similarity metrics (SVCCA, RSA, and TSVDR), UL SNNs consistently achieve the strongest prediction of AM neural responses, outperforming both supervised ANNs and supervised SNNs. Importantly, this correspondence emerges without any region-specific tuning or supervision. As illustrated in Figure 2, UL SNNs exhibit the highest peak neural predictivity and maintain relatively stable performance across network depth, suggesting that their learned representations are both distributed and robust across representational depth.

The superior alignment of UL SNNs with the AM face patch indicates that unsupervised spiking learning can give rise to high-level object representations that closely resemble those formed in primate vision through natural experience. This finding supports the hypothesis that cortical specialization for identity recognition may emerge without explicit category supervision, relying instead on self-organizing mechanisms driven by sensory statistics.

In contrast, supervised ANN models trained with fixed classification objectives show weaker correspondence in deeper layers. This divergence likely reflects the tendency of task-driven labels to impose representational constraints that favor category discrimination over more behaviorally versatile visual information. Such observations align with a growing perspective in neuroscience that biological visual learning prioritizes generalizable representations rather than category-specific optimization (Lin et al., 2017; Chang et al.,

*Table 7.* Downstream task performance under few-shot and noisy transfer settings (Top-1 accuracy, %).

| Tasks | Methods | Task Ratios | | | |
|---|---|---|---|---|---|
| | | 20% | 40% | 60% | 80% |
| Few-shot | SL ANN | 57.14 | 58.19 | 58.20 | 58.27 |
| | UL ANN | 58.59 | 59.79 | 60.13 | 62.84 |
| | SL SNN | 61.49 | 62.95 | 64.04 | 65.20 |
| | **UL SNN** | 62.94 | 64.50 | 65.13 | 65.90 |
| | **Pr. UL SNN** | **63.11** | **64.70** | **65.75** | **66.32** |
| Gaussian Noise | SL ANN | 52.53 | 51.48 | 50.51 | 48.65 |
| | UL ANN | 58.11 | 57.13 | 55.32 | **53.07** |
| | SL SNN | 53.22 | 52.88 | 50.92 | 48.78 |
| | **UL SNN** | 59.07 | 57.52 | 55.42 | 52.44 |
| | **Pr. UL SNN** | **59.94** | **57.69** | **55.60** | 52.57 |
| Salt-and-Pepper Noise | SL ANN | 53.53 | 48.57 | 43.41 | 38.57 |
| | UL ANN | 55.08 | 49.39 | 43.79 | 39.93 |
| | SL SNN | 54.59 | 49.47 | 44.22 | 38.53 |
| | **UL SNN** | 56.98 | 50.11 | 44.45 | 40.02 |
| | **Pr. UL SNN** | **57.27** | **50.49** | **45.20** | **40.24** |

2021; Hong et al., 2016).

### 4.3. Ventral Visual Stream Layer–Area Correspondence

We further examine how hierarchical representations learned by different models align with neural responses along the macaque ventral visual stream, spanning primary visual cortex (V1), intermediate area V4, and inferotemporal cortex (IT). Neural predictivity is evaluated layer by layer and visualized using heatmaps, where darker colors indicate stronger correspondence between model activations and cortical responses.

Among all evaluated model classes in Figure 3, UL SNN exhibits the clearest and most biologically meaningful layer–area correspondence. Specifically, its early layers achieve the highest neural predictivity for V1, intermediate layers align most strongly with V4, and its deepest layers correspond best to IT cortex. This sequential alignment closely mirrors the canonical organization of the primate ventral visual stream, in which low-level visual features are processed in early cortex and progressively transformed into abstract, object-level representations in downstream regions (Carandini et al., 2005; Yamane et al., 2008).

Other models show less coherent hierarchical structure. SL SNN and UL ANN exhibit partial layer–area correspondence, but with weaker transitions and lower peak alignment across cortical areas. SL ANN displays the least distinct hierarchical mapping, with high predictivity for early visual areas persisting across many layers and limited separation between representations corresponding to different cortical stages.

These results indicate that UL SNN does not merely perform well on representational benchmarks; it also internalizes a hierarchical organization that closely parallels the computational structure of primate vision. Importantly, this layer-wise specialization emerges spontaneously, without supervised labels, region-specific supervision, or architectural tuning. Together, these findings suggest that unsupervised sensory learning combined with spiking dynamics is sufficient to drive the emergence of ventral stream–like structure and function. This observation resonates with developmental neuroscience evidence that infants acquire robust visual representations primarily through passive exposure to natural sensory input rather than explicit category instruction (Bergelson & Swingley, 2012; Zhuang et al., 2021).

### 4.4. Functional Alignment

To assess the functional utility of learned representations beyond neural predictivity, we evaluate model performance on downstream tasks that probe behavioral properties of biological vision with a ResNet-18 backbone. Specifically, we consider two scenarios: (1) *rapid learning from limited supervision* (few-shot transfer), and (2) *robust recognition under degraded sensory input* (noisy transfer). In all cases, representations are frozen and evaluated using a linear classifier, ensuring that performance reflects the intrinsic structure and linear decodability of the learned features rather than task-specific fine-tuning.

Representations are pretrained on Tiny ImageNet and transferred to CIFAR-10. Few-shot learning is evaluated using 20%, 40%, 60%, or 80% of labeled training data. For robustness testing, both training and evaluation images are perturbed with Gaussian or Salt-and-Pepper noise at corresponding ratios of 20%–80%.

Across all settings, unsupervised spiking models consistently outperform supervised and non-spiking baselines, with the pruned variant (Pr. UL SNN) achieving the strongest performance.

**Few-shot learning.** As shown in Table 7, Pr. UL SNN maintains the highest Top-1 accuracy at every supervision level. For example, with only 20% labeled data, it achieves 63.11%, exceeding SL SNN (61.49%) and UL ANN (58.59%). This advantage persists as the amount of labeled data increases, indicating that spiking-driven representations are highly linearly decodable and support rapid task adaptation from limited supervision.

**Robustness to noise.** As shown in Table 7, Pr. UL SNN also demonstrates superior robustness under both Gaussian and Salt-and-Pepper noise. Under Gaussian noise with 20% supervision, it achieves 59.94%, outperforming ANN-based models by a clear margin, and this advantage remains at higher noise levels. Similar trends are observed under Salt-and-Pepper noise, where Pr. UL SNN sustains higher accuracy as corruption increases.

The ability to learn efficiently from sparse supervision and

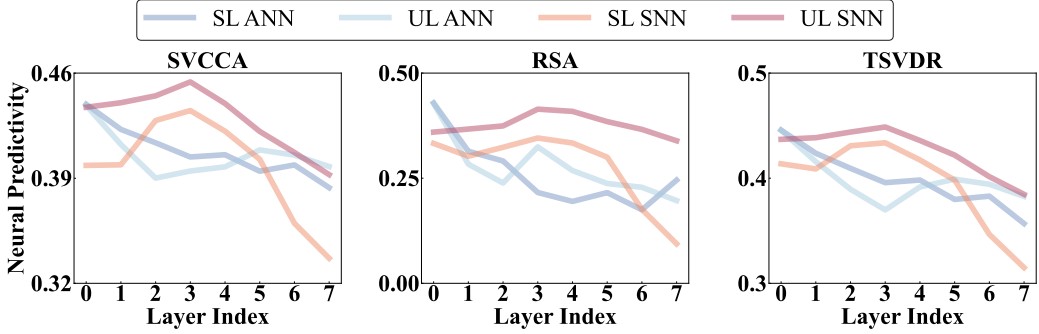

*Figure 2.* Layer-wise neural predictivity for the macaque AM face patch across different model architectures.

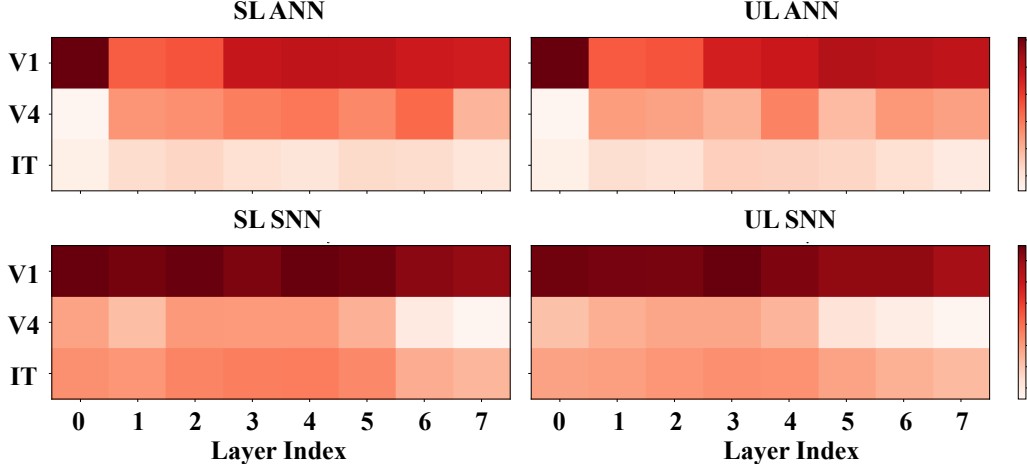

*Figure 3.* Layer-wise neural predictivity across V1, V4, and IT regions of the macaque ventral visual stream for different model classes.

to remain robust under sensory degradation reflects key functional properties of biological vision. The observed improvements suggest that unsupervised learning combined with spiking temporal dynamics yields representations that are both efficient and noise-tolerant. Synaptic pruning further amplifies these benefits by reducing redundancy while preserving task-relevant structure. Together, these results indicate that unsupervised spiking architectures capture not only neural representational alignment but also functional characteristics of the ventral visual pathway, supporting their use as mechanistic models of cortical computation.

### 4.5. Ablation Studies

To disentangle the roles of learning paradigm, neuronal dynamics, and structural refinement, we conduct ablations along three orthogonal dimensions: supervised versus unsupervised learning (SL vs. UL), artificial versus spiking architectures (ANN vs. SNN), and unsupervised SNNs with and without synaptic pruning (UL SNN vs. Pr. UL SNN). Results are summarized in Tables 1–6.

**Effect of unsupervised learning.** Across datasets and

pretraining regimes, unsupervised learning consistently matches or improves neural predictivity relative to supervised training, with the largest gains observed under representational metrics such as RSA and TSVDR. These improvements are especially pronounced in spiking networks: for example, on the Allen Brain Mouse dataset, UL SNN substantially outperforms both SL ANN and UL ANN in RSA, and pruning further enhances this advantage. Similar trends are observed on the Macaque Face dataset, where unsupervised spiking models achieve the strongest TSVDR scores. These results indicate that experience-driven objectives suffice to shape brain-aligned representations under label-sparse conditions.

**Effect of spiking architectures.** Comparisons between architecture-matched ANNs and SNNs reveal a consistent advantage for spiking models, particularly in the unsupervised regime. Across mouse and macaque datasets, UL SNNs achieve higher neural predictivity than UL ANNs, indicating that spike-based temporal dynamics and event-driven computation contribute meaningfully to alignment with cortical activity. While supervised SNNs can be competitive in some settings, their advantages are less consistent

than those observed for unsupervised spiking models.

**Effect of synaptic pruning.** Incorporating activity-dependent synaptic pruning further refines neural alignment on top of unsupervised spiking representations. Across all datasets and metrics, Pr. UL SNN consistently improves upon UL SNN, for example increasing RSA on the Allen Brain Mouse dataset and TSVDR on the Macaque Face and Synthetic datasets under both CIFAR-10 and Tiny-ImageNet pretraining. Rather than introducing qualitatively new structure, pruning amplifies and stabilizes representations already shaped by unsupervised spiking learning, consistent with developmental synaptic refinement.

Overall, these ablation results demonstrate that unsupervised learning, spiking neuronal dynamics, and activity-dependent structural refinement play complementary roles in shaping brain-aligned visual representations. Their combination yields models that are both quantitatively closer to neural data and mechanistically consistent with principles of cortical development and computation.

## 5. Discussion

### 5.1. Emergence of Brain-Aligned Representations

Our results show that biologically aligned visual representations can emerge in deep spiking neural networks through unsupervised sensory exposure and activity-dependent synaptic refinement, without explicit task supervision. Across mouse and macaque datasets, unsupervised SNNs consistently achieve higher neural predictivity than supervised ANN and SNN baselines, while pruning further improves alignment and robustness. These findings suggest that spiking temporal dynamics and structural refinement jointly contribute to cortex-aligned hierarchical representations.

### 5.2. Developmental Interpretation of Synaptic Pruning

This work also provides a developmental interpretation of recent observations that high-dimensional networks can exhibit substantial cortical alignment even without supervised optimization. We propose that synaptic pruning acts as a biologically grounded mechanism for refining latent and over-complete representations into compact and stable functional structure. From this perspective, brain alignment emerges from the interaction between high-dimensional architectures, unsupervised experience, and activity-dependent refinement.

### 5.3. Architectural Generality and Inductive Bias

ResNet is adopted as a controlled backbone to isolate the effects of unsupervised learning, spiking dynamics, and prun-

*Table 8.* Neural predictivity (RSA) across different architectures on the Allen Brain Mouse dataset. Results not labeled "This work" are from (Huang et al., 2023).

| Model Type | Model | RSA Score |
|---|---|---|
| SNN (This work) | **Pr. UL SNN** | **0.4608** |
| SNN (This work) | UL SNN | 0.4279 |
| ANN (CNN) | SqueezeNet1.1 | 0.4360 |
| ANN (CNN) | CORNet-Z | 0.3878 |
| ANN (CNN) | ResNet50 | 0.3200 |
| ANN (CNN) | DenseNet121 | 0.2954 |
| Transformer-like | ConvNeXt-Base | 0.2664 |
| Transformer-like | ConvNeXt-Large | 0.2384 |

ing under a consistent architectural setting. Prior studies suggest that convolutional inductive biases, including locality and hierarchical feature composition, promote cortex-aligned representations.

To further examine generality, we compare representative CNN, SNN, and Transformer-like architectures under RSA evaluation on the Allen Brain Mouse dataset. As shown in Table 8, the proposed Pr. UL SNN achieves the highest neural similarity across all compared model families. These results suggest that the observed improvements primarily arise from the proposed learning and pruning mechanism rather than dependence on a specific backbone architecture.

## Acknowledgements

This work was partly supported by grants from the National Natural Science Foundation of China (62336007, U25D9015), the Starry Night Science Fund of Zhejiang University Shanghai Institute for Advanced Study (SN-ZJU-SIAS-002) ), and the Fundamental Research Funds for the Central Universities.

## Impact Statement

This paper presents work whose goal is to advance the field of machine learning. There are many potential societal consequences of our work, none of which we feel must be specifically highlighted here.

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

# Appendix

# A. Additional Theoretical Analysis

This appendix provides theoretical support for two key design choices in our framework: (1) the use of cosine-similarity-guided pruning to preserve representational geometry, and (2) the use of temporally averaged spiking representations in contrastive learning.

## A.1. Proposition 1: Direction-Preserving Pruning via Cosine Similarity

Let $\omega \in \mathbb{R}^N$ denote a synaptic weight vector, and let $\omega^{(n)}$ denote the vector obtained by setting its $n$-th component to zero, that is,

$$\omega^{(n)} = (\omega_1, \ldots, \omega_{n-1}, 0, \omega_{n+1}, \ldots, \omega_N). \tag{5}$$

The cosine similarity between the original vector $\omega$ and the pruned vector $\omega^{(n)}$ is given by

$$\cos(\omega, \omega^{(n)}) = \frac{\langle \omega, \omega^{(n)} \rangle}{\|\omega\|_2 \, \|\omega^{(n)}\|_2} = \frac{\|\omega^{(n)}\|_2}{\|\omega\|_2}, \tag{6}$$

where $\| \cdot \|_2$ denotes the $\ell_2$ norm.

Since $\|\omega\|_2$ is constant, maximizing the cosine similarity is equivalent to maximizing $\|\omega^{(n)}\|_2$. Noting that

$$\|\omega^{(n)}\|_2^2 = \|\omega\|_2^2 - \omega_n^2, \tag{7}$$

it follows directly that the cosine similarity is maximized by choosing the index $n$ that minimizes $\omega_n^2$. Equivalently, pruning the synapse with the smallest absolute weight preserves the direction of the weight vector to the greatest extent.

This result provides a principled justification for magnitude-based pruning under a cosine similarity criterion, ensuring minimal distortion of representational geometry when removing redundant or weak synaptic connections. While this analysis considers single-step pruning, it motivates the use of cosine-similarity-guided pruning as a direction-preserving heuristic in high-dimensional parameter spaces.

## A.2. Proposition 2: Temporal Averaging of Spiking Representations

Let $\{o(t)\}_{t=1}^T$ denote the spike train of a neuron over $T$ discrete timesteps, where $o(t) \in \{0, 1\}$ indicates whether a spike occurs at time $t$. Define the temporally averaged representation

$$z = \frac{1}{T} \sum_{t=1}^T o(t). \tag{8}$$

Under standard assumptions of rate coding in spiking neurons, the expected value of $z$ provides an unbiased estimate of the neuron's firing rate. Consequently, temporally averaged spiking activity yields a continuous representation that captures the neuron's response intensity while retaining sensitivity to stimulus-driven variations encoded in spike timing.

When contrastive objectives are defined on such temporally averaged representations, the resulting loss approximates conventional rate-based contrastive learning objectives. This establishes a direct connection between spike-timing dynamics and standard representation learning frameworks, and provides theoretical support for applying contrastive learning to spiking neural networks using temporal aggregation.

Together, these two propositions clarify how cosine-similarity-guided pruning preserves representational structure, and how temporally averaged spiking activity enables effective self-supervised representation learning in spiking neural networks.

# B. Implementation Details

## B.1. Pretraining Datasets

Our base models are pretrained on two image datasets: **(1) CIFAR-10** (Krizhevsky et al., 2009) is composed of 50,000 training and 10,000 test images, each of size $32 \times 32$ pixels and labeled into one of 10 object categories. This dataset is

widely used for evaluating image classification models. **(2) Tiny-ImageNet** (Le & Yang, 2015) is a subset of the ImageNet dataset (Krizhevsky et al., 2017), containing 200 classes with 500 training images, 50 validation images, and 50 test images per class. Each image is resized to 64×64 pixels. Tiny-ImageNet presents a more challenging setting than CIFAR-10 due to its larger number of classes and higher visual diversity.

### B.2. Neural Datasets

After pretraining, we evaluate the neural predictivity of our base models using three biological visual neural datasets: **(1) Allen Brain Mouse** (Siegle et al., 2021), which contains recordings from the mouse visual cortex in response to 118 natural scene stimuli, covering six brain regions: VISp, VISl, VISrl, VISal, VISpm, and VISam. **(2) Macaque Face** (Chang et al., 2021), which includes recordings from 159 neurons in the anterior medial (AM) face patch of the macaque visual cortex in response to 2,100 real face stimuli. **(3) Macaque Synthetic** (Majaj et al., 2015), which consists of recordings from 88 V4 neurons and 168 IT neurons in the macaque visual cortex in response to 3,200 synthetic image stimuli. In addition, we preprocess all neural datasets following the same procedure described in (Huang et al., 2023).

