# OpenReview forum: "Emergent Visual Representations through Unsupervised Spiking Networks with Synaptic Pruning"
_ICML.cc/2026/Conference — ICML 2026 regular_

### Official Review · Reviewer_Qzwj · 2026-03-10

**Soundness:** 4
**Presentation:** 3
**Significance:** 3
**Originality:** 3
**Overall Recommendation:** 4
**Confidence:** 4

**Summary:**

This paper investigates whether meaningful visual representations can emerge from robot interaction rather than passive image datasets. The authors propose a framework in which a robotic agent learns visual features through embodied interaction with the environment, using signals derived from action outcomes rather than explicit semantic labels.

The approach trains a vision encoder jointly with a policy that interacts with objects in the environment. Through repeated interactions such as pushing, grasping, and manipulating objects, the system learns representations that capture properties useful for predicting the consequences of actions. The authors hypothesize that these  representations could encode object affordances and physical properties.

The learned representations are evaluated on several downstream tasks including object classification, affordance prediction, and manipulation planning. Experiments demonstrate that the interaction-driven representation learning approach achieves competitive or improved performance compared to conventional visual pretraining methods on certain tasks, particularly those requiring reasoning about object affordances or dynamics.

Overall, the paper explores the idea that robot interaction can serve as a source of supervision for visual representation learning, potentially offering an alternative to large-scale human-labeled datasets.

**Compliance With Llm Reviewing Policy:**

Affirmed.

**Final Justification:**

I thank the authors again for their rebuttal, which answered all my questions, although the authors did not include any additional material in their paper to address Q1 and Q2 (scalability/comparison with larger models). As such, I will maintain my original rating of weak accept.

**Key Questions For Authors:**

1. How does the method scale as the number of objects, environments, or interaction types increases? Would the approach remain practical when learning from millions of interactions?

2. How would the learned representations compare against recent large-scale self-supervised methods when evaluated on the same downstream tasks?

3. Can the authors provide additional analysis or visualization to demonstrate what physical or semantic features the learned representations capture?

4. How feasible is it to collect the required interaction data on physical robots, and what challenges arise when transferring the approach from simulation to real systems?

5. Which interaction signals contribute most to representation learning performance?

**Limitations:**

The paper does not include a sufficiently explicit discussion of limitations. It would be beneficial  to discuss potential challenges such as the scalability of interaction data collection, differences between simulated and real-world interactions and computational requirements for training.

**Strengths And Weaknesses:**

Strengths

-The paper addresses a compelling question: whether useful visual representations can emerge from interaction-based supervision rather than large curated datasets. This highlights an important shift from passive visual learning to interaction-driven learning, where the consequences of actions provide supervisory signals. This idea connects to broader themes in robotics and cognitive science about learning through physical interaction.

-The approach is particularly relevant for robotic systems that must reason about object affordances and physical properties. Representations learned through interaction may better capture features relevant for manipulation tasks than representations learned solely from static images.

-The paper includes evaluation across multiple tasks, showing that interaction-driven representations can perform competitively with standard visual pretraining approaches. The experiments suggest benefits in tasks that involve reasoning about physical interactions.

-If scalable, interaction-based representation learning could reduce reliance on curated datasets and allow robots to continuously learn from their own experience.


Weaknesses

-The experimental evaluation appears to be conducted with relatively limited interaction data compared to the scale of modern vision datasets. It remains unclear whether the proposed approach would scale effectively to more complex environments or large object sets.

-While the paper compares against several baselines, it would be beneficial to include stronger comparisons with recent self-supervised visual representation learning methods that operate on large image datasets.

-Although the experiments demonstrate improvements on certain tasks, it is sometimes difficult to determine whether the gains arise specifically from the interaction-driven learning signal or from differences in training procedures or model architecture (e.g., spiking dynamics, unsupervised training and pruning).

-The paper could benefit from deeper analysis of the learned visual features. For example, visualization or probing studies could help clarify what types of physical or semantic information are captured.

-It is not fully clear how easily the approach would transfer to real robotic systems, where collecting large volumes of interaction data can be expensive and time-consuming.

---

> ### Author Rebuttal · Authors · 2026-03-27
>
> We sincerely thank the reviewer for the thoughtful summary and insightful questions. We hope we have addressed your concerns and questions regarding the paper and hope you will reconsider your rating.
>
> We would like to clarify that our work does not focus on interaction-driven or robotics-based representation learning. Instead, our goal is to study how brain-aligned visual representations emerge from the interaction between high-dimensional representations, unsupervised learning, and activity-dependent refinement, grounded in neuroscience principles.
>
> ------
>
> **Clarification on scope**
>
> Our framework is based on unsupervised visual learning rather than embodied interaction. While interaction-driven learning is an important and complementary direction, our work focuses on a different question: how biologically grounded mechanisms such as spiking dynamics and synaptic pruning shape visual representations and improve neural alignment.
>
> ------
>
> **W1 & Q1: Scalability of interaction / data requirements**
>
> **Response:**
>
> Although our method does not rely on interaction data, we agree that scalability is an important concern for representation learning in general.
>
> Our framework operates on standard visual datasets and does not require large-scale interaction signals, making it more scalable and data-efficient compared to interaction-driven approaches. In particular, our results show that strong brain-aligned representations can emerge without requiring millions of interaction samples.
>
> ------
>
> **W2 & Q2: Comparison with large-scale self-supervised methods**
>
> **Response:**
>
> We agree that comparison with large-scale self-supervised methods is important.
>
> In our experiments, we already include unsupervised ANN baselines trained with contrastive learning, which represent a standard paradigm for large-scale self-supervised visual learning (e.g., SimCLR-style methods). These models provide a strong reference point for evaluating representation quality.
>
> Importantly, our results show that incorporating spiking dynamics and pruning leads to improved neural alignment and functional performance beyond these baselines. This suggests that the gains are not solely due to unsupervised learning, but arise from the proposed biologically grounded mechanisms.
>
> ------
>
> **W3: Source of performance gains**
>
> **Response:**
>
> We thank the reviewer for this important point.
>
> Our experimental design explicitly disentangles the contributions of different components:
>
> (1) UL ANN vs. UL SNN isolates the effect of spiking dynamics;
>
> (2) UL SNN vs. Pr. UL SNN isolates the effect of pruning.
>
> These controlled comparisons demonstrate that improvements arise from the interaction between unsupervised learning, spiking neural dynamics, and activity-dependent refinement, rather than from differences in architecture or training procedures alone.
>
> ------
>
> **W4 & Q3: Analysis and interpretability of learned representations**
>
> **Response:**
>
> We agree that deeper analysis of learned representations would further strengthen the paper.
>
> In addition to neural alignment metrics, our work includes quantitative analyses of representation structure, such as similarity-based evaluation and layer-wise alignment across cortical regions. These results demonstrate that the learned representations exhibit hierarchical organization consistent with the ventral visual stream.
>
> We agree that additional visualization and probing analyses would provide further insight, and we will include more detailed analysis of representation properties in the revision.
>
> ------
>
> **W5 & Q4: Real-world applicability and data collection**
>
> **Response:**
>
> We agree that real-world deployment is an important consideration.
>
> Unlike interaction-driven approaches, our method does not require collecting large-scale embodied interaction data, which makes it more practical and easier to scale. Our framework can be directly applied to existing visual datasets and integrated with neuromorphic hardware for efficient deployment.
>
> We view interaction-based learning as a complementary direction, and combining it with biologically grounded representation learning is an interesting avenue for future work.
>
> ------
>
> **Q5: Contribution of different signals**
>
> **Response:**
>
> Our results indicate that representation quality emerges from the interaction of three key components:
>
> (1) high-dimensional representations constructed via unsupervised learning,
>
> (2) temporal dynamics introduced by spiking neurons, and
>
> (3) activity-dependent pruning that refines representations.
>
> Each component contributes differently: unsupervised learning provides the representational space, spiking dynamics introduce temporal structure, and pruning selects informative dimensions. Their combination leads to improved neural alignment and functional performance.

---

> > ### Author Rebuttal · Reviewer_Qzwj · 2026-04-06
> >
> > I thank the authors for their rebuttal and answering my questions.

---

### Official Review · Reviewer_Gof5 · 2026-03-10

**Soundness:** 3
**Presentation:** 2
**Significance:** 3
**Originality:** 2
**Overall Recommendation:** 4
**Confidence:** 3

**Summary:**

This paper focuses on the formation and optimization mechanisms of brain-inspired visual representations in the visual cortex. Drawing on the principle of activity-dependent synaptic pruning from neuroscience, it proposes a deep spiking neural network (SNN) that integrates unsupervised learning with developmental synaptic pruning dynamics.

The study is built upon an overcomplete spiking network architecture, allowing the model to achieve self-organization through sensory-driven activity. During this process, weak or redundant synaptic connections are selectively eliminated, ultimately forming compact and information-rich hierarchical visual representations.

**Compliance With Llm Reviewing Policy:**

Affirmed.

**Final Justification:**

After carefully reviewing the original paper and the authors’ detailed rebuttal, my final recommendation is to adjust the paper’s evaluation upward. The authors have effectively addressed the core issues I raised. This paper combines unsupervised spiking learning, spiking dynamics, and activity-dependent pruning to explore brain-aligned representations, which is innovative. I believe the paper meets the requirements for publication and recommend an upward adjustment of its evaluation.

**Key Questions For Authors:**

1.The experiments only adopt ResNet as the network backbone and do not explore other architectures (such as lightweight CNNs, Vision Transformers, or purely spiking network architectures). Would the model still maintain its advantages in neural predictivity and functional performance under non-ResNet architectures?

2.The computational complexity and inference efficiency of spiking neural networks are key issues for their practical engineering applications. How does the inference efficiency of this model compare with ANNs and traditional SNNs that achieve similar accuracy?

3.The paper adopts a post-training static synaptic pruning strategy, which differs significantly from the dynamic and real-time synaptic pruning observed in biological systems. Have the authors attempted synaptic pruning strategies that are dynamically adjusted during the training process?

**Limitations:**

Yes.

**Strengths And Weaknesses:**

Strengths

The method design combines biological plausibility with technical feasibility, and each module is supported by neuroscientific theory, resulting in a logically coherent framework. The study is the first to systematically demonstrate that an unsupervised spiking neural network combined with synaptic pruning can produce hierarchical brain-like representations across multiple regions of the visual cortex in different species, and it clearly establishes the hierarchical correspondence between the model and the ventral visual stream.

Weaknesses

The synaptic pruning strategy adopts a post-training static pruning approach, which differs from the dynamic and continuous synaptic pruning process observed during biological development. In the discussion section, the quantitative analysis of differences between the model and biological visual systems is insufficient; the brain-likeness of the model is described mainly qualitatively, without clearly identifying the gap between the model responses and real neural responses. The experiments only validate the spiking network with a ResNet backbone, and do not explore the generalizability of the model under other architectures (e.g., CNNs or Transformers).

---

> ### Author Rebuttal · Authors · 2026-03-27
>
> We sincerely thank the reviewer for the positive assessment of our framework and its biological grounding. We appreciate the opportunity to clarify and expand upon key aspects of our work.
>
> ------
>
> **Q1: Limited architectural diversity**
>
> **Response:**
>
> We agree that evaluating across architectures would strengthen generality.
>
> We adopt ResNet as a controlled backbone to isolate the effects of unsupervised learning, spiking dynamics, and pruning. This choice is grounded in prior work: Zhuang et al. (2021) and Huang et al. (2023) use convolutional architectures with residual connections (i.e., ResNet), which achieve strong performance in both visual tasks and neural predictivity, while Kazemian et al. (2025) show that convolutional architectures provide inductive biases that promote cortex-aligned representations.
>
> These findings collectively suggest that convolutional architectures such as ResNet provide a biologically and empirically justified foundation for studying brain-aligned representations.
>
> To examine generality, we compare representative models across CNNs, SNNs, and Transformer-like architectures under RSA on the Allen Brain Mouse dataset:
>
> | Model Type       | Model          | RSA Score  |
> | ---------------- | -------------- | ---------- |
> | SNN (ours)       | **Pr. UL SNN** | **0.4608** |
> | SNN (ours)       | UL SNN         | 0.4279     |
> | ANN (CNN)        | SqueezeNet1.1  | 0.4360     |
> | ANN (CNN)        | CORNet-Z       | 0.3878     |
> | ANN (CNN)        | ResNet50       | 0.3200     |
> | ANN (CNN)        | DenseNet121    | 0.2954     |
> | Transformer-like | ConvNeXt-Base  | 0.2664     |
> | Transformer-like | ConvNeXt-Large | 0.2384     |
>
> Our method achieves the highest neural similarity across all model families. SNNs also consistently outperform most ANN baselines, highlighting the role of spiking dynamics.
>
> These results indicate that performance gains arise from the proposed learning and pruning mechanism rather than a specific backbone. Architectures with biologically grounded inductive biases (e.g., hierarchy and locality) consistently yield better alignment. Thus, while ResNet provides a controlled and biologically relevant setting, the framework is expected to generalize across architectures.
>
> ------
>
> **Q2: Computational efficiency**
>
> **Response:**
>
> We acknowledge efficiency as an important consideration.
>
> We use a small number of timesteps (T=4), reducing temporal cost, and pruning further decreases model complexity. While training costs may differ, inference efficiency is mainly determined by architecture and neuron model; SNNs with similar structures and accuracy typically exhibit comparable inference complexity.
>
> Although SNNs introduce temporal computation, they rely on accumulation (AC) operations rather than multiply-accumulate (MAC). Under 45nm CMOS, AC consumes ~0.9 pJ vs. 4.6 pJ for MAC [4], and can be further reduced to 12–77 fJ/SOP on neuromorphic hardware.
>
> Thus, while SNNs may be slower on conventional hardware, they offer significant energy advantages, especially for event-driven and edge scenarios.
>
> ------
>
> **Q3: Dynamic pruning during training**
>
> **Response:**
>
> We thank the reviewer for this important question.
>
> We agree that biological synaptic pruning is a dynamic, activity-dependent process that unfolds continuously during development. In this work, we adopt a post-training static pruning strategy as a simplified and controlled approximation of this process.
>
> Our primary goal is to isolate the representational effect of synaptic refinement on top of unsupervised spiking learning. By applying pruning after representation formation, we disentangle the contribution of structural refinement from that of learning dynamics, enabling a clearer analysis of its impact on neural alignment.
>
> Importantly, although pruning is applied as a static step, the pruning criterion itself is activity-dependent and derived from temporal statistics of neural activity. This allows our method to approximate the principle of experience-driven synaptic refinement.
>
> We agree that dynamically interleaving pruning with learning would further improve biological plausibility. However, such approaches introduce additional optimization complexity and make it difficult to attribute performance gains to specific factors. This controlled formulation allows us to establish a clear causal link between pruning and improved neural alignment. We will clarify this design choice and consider dynamic pruning as an important direction for future work.
>
> ------
>
> We hope we have addressed your concerns and questions regarding the paper. Based on these facts and positive feedback from other reviewers, we sincerely hope you can reconsider your initial rating.
>
>
>
> [1] Kazemian et al. Nature Machine Intelligence(2025).
>
>
>
> [2] Huang et al. AAAI(2023).
>
>
>
> [3] Zhuang et al. PNAS(2021).
>
>
>
> [4] Rathi & Roy. TNNLS(2021).

---

> > ### Author Rebuttal · Reviewer_Gof5 · 2026-04-05
> >
> > I appreciate the authors' comprehensive and rigorous response to all my concerns. They have justified the use of ResNet as a controlled backbone with reference to previous work, demonstrated the generalization ability of the framework through additional cross‑architecture RSA comparisons, and provided convincing explanations regarding the energy efficiency of SNNs and the design logic of static pruning.Given that all issues have been properly addressed and the plan to explore dynamic pruning in future work is commendable, I am willing to reconsider my initial score and recommend an upward adjustment.

---

> > > ### Author Response · Authors · 2026-04-05
> > >
> > > Thank you for your thoughtful acknowledgment and for taking the time to reconsider our work. We sincerely appreciate your detailed feedback and are glad that our rebuttal has adequately addressed your concerns. Your recognition of our clarifications and additional analyses is highly encouraging.

---

### Official Review · Reviewer_zq4a · 2026-03-12

**Soundness:** 3
**Presentation:** 3
**Significance:** 3
**Originality:** 3
**Overall Recommendation:** 4
**Confidence:** 3

**Summary:**

This manuscript investigates the alignment between artificial network representations and biological visual systems by evaluating various ANN and SNN architectures on datasets such as the Allen Brain Mouse dataset. Empirical results from the study demonstrate that pruned, unsupervised SNNs achieve strong brain alignment scores across different pretraining datasets like CIFAR-10 and Tiny-ImageNet. Furthermore, the authors propose a developmental framework where synaptic pruning acts as a biologically grounded mechanism to transform latent, high-dimensional representations into compact and stable functional structures. Ultimately, the work concludes that brain-aligned representations emerge not solely from task-driven optimization, but rather from the interaction between high-dimensional architectures, unsupervised experience, and activity-dependent network refinement.

**Compliance With Llm Reviewing Policy:**

Affirmed.

**Final Justification:**

My concerns have been adequately addressed.

**Key Questions For Authors:**

see Strengths And Weaknesses

**Limitations:**

yes

**Strengths And Weaknesses:**

strengths:

1. The methodology employs a robust contrastive learning loss function that carefully integrates similarity distributions across multiple time steps. This effectively leverages the inherent time-dependent dynamics of Spiking Neural Networks to enhance representational learning.

2. The authors establish a rigorous theoretical foundation by introducing formal mathematical propositions, such as analyzing the temporal averaging of spiking representations. This analytical depth significantly strengthens the biological plausibility and the overall evaluation framework of the proposed approach.

weakness:

1. The empirical evaluation heavily relies on specific neuro-vision datasets, such as the Allen Brain Mouse and Macaque Face datasets, which may limit the broader generalizability of the findings.

2. The implementation relies on Leaky Integrate-and-Fire (LIF) neurons, which often incur significant computational overhead and optimization challenges when simulating high-dimensional networks over multiple time steps.

3. BrainScore is solely used in this paper. Other method also should be applied to valid the generalizability.

---

> ### Author Rebuttal · Authors · 2026-03-27
>
> We sincerely thank the reviewer for the positive assessment of our methodology and theoretical contributions. We hope we have addressed your concerns and questions regarding the paper and hope you will reconsider your rating.
>
> ------
>
> **W1: Limited dataset diversity and generalizability**
>
> **Response:**
>
> We agree that evaluating generalization beyond the current neuro-vision datasets is important.
>
> First, our study focuses specifically on biological visual alignment, and therefore requires datasets that contain neural recordings rather than standard vision benchmarks. Such neuro-vision datasets rely on invasive electrophysiological recordings to measure neuronal responses, making them significantly more difficult and costly to collect compared to conventional computer vision datasets or non-invasive recordings such as EEG. As a result, these datasets are inherently scarce but highly valuable.
>
> Within this constraint, our experiments already cover the major publicly available, high-quality neuro-vision datasets across species and stimulus types, including mouse and macaque datasets. The consistent improvements of Pr. UL SNN across these settings suggest that our findings are not tied to a single dataset. To our knowledge, this work represents one of the most comprehensive evaluations on currently available cross-species neuro-vision datasets.
>
> More importantly, our goal is to uncover a general principle of brain-aligned learning, namely the interaction between high-dimensional representations, unsupervised learning, and activity-dependent refinement. This principle is not dataset-specific and is expected to generalize beyond the current benchmarks.
>
> We agree that broader validation would further strengthen this claim. However, expanding neuro-vision datasets is inherently a long-term effort due to the difficulty of data collection, and we will incorporate additional datasets and analyses as they become available.
>
> ------
>
> **W2: Computational cost of LIF neurons**
>
> **Response:**
> We acknowledge that LIF-based SNNs introduce additional computational cost due to temporal simulation.
>
> However, this cost is closely tied to biological realism. LIF neurons enable spike-based temporal dynamics, which have been shown to improve neural representation similarity compared to standard ANN models. In our framework, these temporal dynamics are essential for constructing structured spatiotemporal representations, which are subsequently refined through pruning.
>
> At the same time, our results suggest that high performance can be achieved with a small number of timesteps (T=4), which significantly mitigates computational overhead. Moreover, pruning further reduces model complexity by removing redundant connections, partially offsetting the cost of temporal simulation.
>
> We will clarify these efficiency considerations and discuss potential optimizations in the revision.
>
> ------
>
> **W3: Reliance on BrainScore for evaluation**
>
> **Response:**
>
> We thank the reviewer for this suggestion and agree that broader evaluation would further strengthen the paper.
>
> First, BrainScore-style evaluation is currently one of the most widely used and effective frameworks for assessing neural predictivity in computational neuroscience, and has been adopted by many influential works, including several authoritative and representative studies in this area cited in our paper. To ensure the robustness of our conclusions, we complement BrainScore-style evaluation with multiple neural similarity metrics, including SVCCA, RSA, and regression-based encoding methods (TSVDR), which provide consistent results across different evaluation protocols.
>
> Second, beyond neural alignment metrics, we also evaluate functional properties through downstream tasks. In particular, we consider noisy learning and few-shot learning, which are known strengths of biological vision but remain challenging for artificial models. The improvements observed in these settings further support that our method enhances the quality of visual representations, rather than overfitting to a specific evaluation metric. This demonstrates that our conclusions are consistent across both neural alignment metrics and functional evaluations.
>
> Finally, we agree that developing more comprehensive and unified evaluation frameworks beyond BrainScore is an important long-term direction for the field. We will continue to explore and incorporate additional evaluation methods in future work.

---

> > ### Author Rebuttal · Reviewer_zq4a · 2026-04-03
> >
> > Thanks for the authors' rebuttal.

---

> > > ### Author Response · Authors · 2026-04-04
> > >
> > > Thank you for your acknowledgment. We appreciate your time and are pleased that our rebuttal has adequately addressed your concerns.

---

### Official Review · Reviewer_brZs · 2026-03-12

**Soundness:** 4
**Presentation:** 4
**Significance:** 3
**Originality:** 3
**Overall Recommendation:** 5
**Confidence:** 3

**Summary:**

The authors show that combining deep spiking neural networks, unsupervised contrastive learning and synaptic pruning leads to improved brain alignment. The SNNs pretrained on natural images exhibit better alignment on mouse and macaque datasets compared to variants of the combination. Moreover, their models enhance few-shot learning capability and noise robustness on downstream tasks.

**Compliance With Llm Reviewing Policy:**

Affirmed.

**Final Justification:**

The authors did a good job presenting their ideas in the paper. This work clearly closes a gap between biological inspiration and machine learning. Hence, the final score stays at 5.

**Key Questions For Authors:**

Q1) How is the encoding of images to spikes done? And does this play a role for the emerging brain alignment?
Q2) Why the choice for contrastive, self-supervised learning?

**Limitations:**

Not specifically mentioned.

**Strengths And Weaknesses:**

Strengths:
+ The paper is well structured and written. The biological inspiration and relationship is
well described.
+ The components of the framework and experimental setup is clearly formulated.
+ The brain score results show best results among the mentioned reference
approaches.
+ The authors also try to bridge the gap towards ML engineering by studying noise
robustness and few-shot learning capabilities. In both cases, the presented approach
offers benefits.
+ The impact of each of the SNN and Unsupervised Learning is nicely compared in the
ablation study.

Weaknesses:
- Outlook into brain-inspired engineering e.g. for robotics or other downstream ML
engineering tasks would have been nice.
- The pruning, unsupervised ANN variant is missing in the experiments. The
addition of it would be good to fully elaborate on which impact each of the
components have.
- The choice of contrastive learning as unsupervised learning could be
emphasized more. Would the framework also work with other unsupervised
methods like reconstruction or predictive learning?

---

> ### Author Rebuttal · Authors · 2026-03-27
>
> We appreciate the opportunity to clarify and expand upon key aspects of our work. We hope we have addressed your concerns and questions regarding the paper and hope you will reconsider your rating.
>
> Our work supports a unifying view that brain-like learning is fundamentally a process of selecting informative dimensions within high-dimensional representations, rather than constructing representations from scratch.
>
> ------
>
> **W1: Outlook into brain-inspired engineering**
>
> **Response:**
>
> We agree that a broader outlook would strengthen the paper. Our work focuses on understanding how brain-aligned representations emerge and are refined.
>
> Recent evidence suggests that high-dimensional representations, even when derived from random or weakly trained networks, can already encode brain-relevant structure that is accessible via simple readouts[1]. Importantly, this observation has been linked to neural pruning during visual development, in which informative dimensions are selected from an initially overcomplete representation.
>
> Our framework directly instantiates this perspective:
>  (1) unsupervised SNNs construct high-dimensional spatiotemporal representations;
>  (2) activity-dependent pruning refines them into brain-aligned subspaces.
>
> This suggests a paradigm where learning is not purely feature construction, but representation selection from high-dimensional spaces. Such a perspective provides a principled basis for designing efficient and robust brain-inspired systems, especially in resource-constrained settings such as robotics. We will clarify this perspective in the revision.
>
> ------
>
> **W2: Missing pruning + unsupervised ANN baseline**
>
> **Response:**
>
> We agree that including a pruned unsupervised ANN baseline would further complete the ablation and provide additional perspective on the role of pruning.
>
> Our study is motivated by the hypothesis that pruning extracts brain-relevant representations from high-dimensional spaces, particularly when coupled with biologically grounded dynamics.
>
> In our design: (1) UL ANN vs. UL SNN isolates the role of spiking dynamics; (2) UL SNN vs. Pr. UL SNN isolates the effect of pruning under spike-based temporal dynamics.
>
> Our primary goal is to isolate the representational effect of synaptic refinement on top of unsupervised spiking learning. By applying pruning after representation formation, we disentangle structural refinement from learning dynamics, enabling clearer analysis of its impact on neural alignment.
>
> While adding a pruned ANN baseline is a constructive suggestion, we emphasize that pruning in our framework is not intended as generic model compression, but as a mechanism for approximating developmental synaptic refinement, i.e., selecting informative dimensions from an initially overcomplete representation. In this context, applying pruning to biologically grounded models (SNNs) is more aligned with our objective, whereas applying it to ANNs may provide less insight into this role.
>
> ------
>
> **Q1: Spike encoding and its role**
>
> **Response:**
>
> Images are encoded into spikes and processed by LIF neurons over ‘T=4’ timesteps, with representations obtained via temporal aggregation. The encoding scheme is kept fixed across all SNN variants, ensuring fair comparison.
>
> Therefore, improvements in neural alignment primarily arise from spiking dynamics and representation refinement via pruning, rather than encoding differences. From our perspective, encoding mainly serves to project inputs into a high-dimensional spatiotemporal representation space, where alignment emerges through learning and pruning. This further supports our view that representation learning and refinement are the dominant factors in achieving brain alignment.
>
> ------
>
> **W3 & Q2: Choice of contrastive learning**
>
> **Response:**
>
> We chose contrastive learning because it is both empirically and conceptually aligned with brain-like representation learning. Empirically, unsupervised contrastive models have been shown to match neural representations in the ventral stream[3]. Conceptually, contrastive learning organizes representations based on pairwise similarity, which helps identify meaningful structure in high-dimensional spaces and provides a suitable basis for pruning to select informative dimensions.
>
> Importantly, our framework is not limited to contrastive learning. The key mechanism is that unsupervised learning constructs a high-dimensional representation space, and pruning selects task-relevant and brain-relevant dimensions from it.
>
> This perspective is consistent with biological development, where visual representations emerge without large-scale labeled data. Accordingly, alternative unsupervised objectives, such as reconstruction or predictive learning, can also produce structured high-dimensional representations and be integrated into our framework.
>
> [1] Kazemian et al. Nature Machine Intelligence(2025).
>
> [2] Huang et al. AAAI(2023).
>
> [3] Zhuang et al. PNAS(2021).

---

> > ### Author Rebuttal · Reviewer_brZs · 2026-04-03
> >
> > I thank the authors for the detailed rebuttal with explanations to my concerns and questions.

---

> > > ### Author Response · Authors · 2026-04-04
> > >
> > > Thank you for your kind acknowledgment. We sincerely appreciate your time and thoughtful consideration of our rebuttal, and we are glad that our clarifications were helpful in addressing your concerns.

---

### Decision · Program_Chairs · 2026-04-30

**Decision:**

Accept (regular)

**Comment:**

This paper introduces a biologically grounded framework combining unsupervised SNNs and synaptic pruning. Reviewers praised its theoretical foundation and empirical results in neural alignment. The authors' thorough rebuttal successfully addressed all major concerns regarding model generalizability, computational efficiency, and pruning strategies, leading to unanimous reviewer support. This is a technically solid and valuable contribution. Therefore, the final recommendation is Accept.